# Limitations of Spatial Judgment Bias Test Application in Horses (*Equus ferus caballus*)

**DOI:** 10.3390/ani12213014

**Published:** 2022-11-03

**Authors:** Giovanna Marliani, Irene Vannucchi, Irini Kiumurgis, Pier Attilio Accorsi

**Affiliations:** Department of Veterinary Medical Sciences, University of Bologna, Via Tolara di Sopra, 50, 40064 Ozzano dell’Emilia, Italy

**Keywords:** judgment bias test, equine, E-BARQ, cortisol, affective state

## Abstract

**Simple Summary:**

Public awareness about animal welfare is increasing and new research interest in this field is the evaluation of affective states. This research aimed to highlight possible limitations in the application of a spatial judgment bias test for the assessment of affective states in horses, considering the influence of personality traits, stress levels, and the structure of the test. Horses were trained to distinguish between a positive position, where they found a bucket with food reward, and a negative position where the bucket was empty. The training ended when the subjects approached the positive position faster than the negative one. Then, three ambiguous positions (Near Positive, Middle, Near Negative) were presented to them, and the latency to reach these three positions allowed their classification as pessimistic or optimistic. Our results showed that some personality traits can influence horses’ responses to ambiguous cues. In addition, the spatial nature of such a test seems inappropriate for horses, which use lateralization when evaluating new objects. Therefore, this preliminary study suggested that this type of test should be modified considering species-specific and individual peculiarities.

**Abstract:**

Affective states are of increasing interest in the assessment of animal welfare. This research aimed to evaluate the possible limitations in the application of a spatial judgment bias test (JBT) in horses, considering the influence of stress level, personality traits, and the possible bias due to the test structure itself. The distinction between two positions, one rewarded (Positive) and the other not (Negative), was learned by 10 horses and 4 ponies,. Then, the latency to reach three unrewarded ambiguous positions (Near Positive, Middle, Near Negative) was measured. Furthermore, the validated Equine Behavior Assessment and Research Questionnaire (E-BARQ) was employed to assess personality traits. Fecal and hair cortisol levels were measured through radioimmunoassay (RIA), and the frequency of behavioral stress indicators was recorded. Results showed that horses that had the rewarded position (Positive) on the right approached Near Negative and Middle faster than those that had Positive on the left. Certain personality traits influenced the latency to reach Middle and Near Positive, but chronic stress did not seem to affect horses’ judgment bias. This preliminary study highlighted several limitations in the employment of spatial JBT for the assessment of affective state in horses and that personality traits can partially influence the cognitive process. Further research is needed to refine the use of this test in horses, considering the peculiarities both of species and of individuals.

## 1. Introduction

In the discipline of animal welfare science, affective states of animals are becoming of increasing interest [1]. The Treaty of Amsterdam on the Protection and Welfare of Animals officially recognized them as sentient beings [2]. In addition, the Five Domain Model for animal welfare evaluation has introduced the “mental domain”, as a result of affective states inferred by external circumstances (environment, nutrition, health, and behavioral interactions) [3].

Animal sciences are increasingly focusing on the exploration of affective states. Emotional experiences are characterized by valence and a state of activation or arousal. In animals, emotional states arise in response to a potential reward that enhances fitness or punishment that threatens fitness. The arousal and valence of emotions are determined by the success or failure of the animal in acquiring a reward or avoiding a punishment [4]. In particular, it is possible to recognize discrete emotions, which are short-lasting and dependent on a certain stimulus, as well as long-lasting emotional states (moods), which are indicative of the environment where the subject lives in terms of possibilities of punishments and rewards and how it copes with those [4]. Often the emotions of animals are investigated using behavioral tests, usually assessing anxiety or fear, and physiological parameters [5,6]. For example, the activation of the hypothalamic–pituitary–adrenal (HPA) axis, which can occur in different situations, can be a good indicator of emotional arousal, but it does not give suggestions about the valence of the underlying emotion (punishing or rewarding) [6]. Therefore, to solve this problem, physiological indicators are considered together with behavioral measurements, but sometimes these parameters can be incongruent and their interpretation can be difficult [4,5]. In addition, often the physiological and behavioral response of an individual facing certain stimuli used in a test can vary according to its subjective experience and mood. Consequently, in non-human animals, the investigation of long-lasting emotional states is challenging, but human psychology gives good solutions to obtain an approximation for it. In particular, it has long been recognized that cognition and emotional state, especially mood, influence each other, and thus cognitive processes, such as memory, decision-making, and attention can be useful tools in the assessment of emotions [4,5,7]. Considering the hypothesis that the effect of emotions on cognitive function has an adaptive value [8,9], the exploration of cognitive indicators of emotional states also becomes valuable for animals [7]. Indeed, different emotions arise in response to threats and opportunities and they influence thoughts and actions in response to them. For example, an individual that is in a state of fear or anxiety will be more prone to negatively judge a neutral stimulus or to better memorize a threatening situation [3,8].

In recent decades, many studies employed cognitive tests to investigate emotions in animals. The most popular is the judgment bias test (JBT), which is based on the decision-making process when facing ambiguous cues [10]. The JBT includes a period of training, during which animals learn to distinguish between two stimuli, one with positive valence and the other with a negative one. Once the animals have learned the difference, they are presented with ambiguous stimuli [11]. The animals that respond as if they expect a negative outcome from the ambiguous cues are considered to have a negative affective state, while animals that have positive expectations and behave accordingly are presumed to be in a positive emotional state. According to their operational response, the individuals can be defined as “pessimistic” or “optimistic” [12]. However, the JBT results can be indicative of a long-term affective state, but there can be other influential factors such as recent positive or negative experiences, particular training conditions, and the nature of the task [10,13]. In addition, there is a strong relationship between personality and emotional states. Stable personality traits can affect mood and this can influence the decision-making process and JBT results [14,15,16,17]. For example, proactive pigs seem to have an optimistic judgment of ambiguous cues [16], while fearful and inactive heifers are more pessimistic than inactive non-fearful heifers [15]. In addition, dogs scoring higher in negative personality traits (fear, aggression, and anxiety) are more likely to judge an ambiguous stimulus negatively, while dogs with higher sociability scores are more optimistic [14].

In a recent review, Hausberger and colleagues [18] indicate the horse as a potential animal model for the study of interaction between welfare conditions and cognition. Moreover, several studies used horses as a model to be studied under stressful conditions, such as dietary challenges [19,20,21]. However, the research of the JBT in horses broadened only recently, especially as regards the effect of the environment and training method on the affective state of this species [22,23,24,25]. Nevertheless, further research is warranted because of the lack of a standardized protocol and insufficient studies on the influence of personality traits on JBT results in horses. Other studies have evidenced how several factors, such as lateralization and temperament, can influence test results in horses [26]. Sensorial lateralization, for example, emerges when they have to evaluate situations with different emotional valences [27,28]. Therefore, these aspects should be taken into account when we want to standardize a test protocol in this species.

This research aimed to investigate the limitations of the application of a spatial JBT in horses, considering the influence of their stress level and personality traits, and the test structure itself. For this, we explored the acute and chronic activation of the HPA axis (by measuring fecal and horsehair cortisol levels [29,30]) and assessed the horses’ personalities (by the Equine Behavior Assessment and Research Questionnaire—E-BARQ [31]). We hypothesized that both acute stress, potentially caused by the test procedure itself, and individual chronic stress could influence the horses’ responses during the JBT, causing a negative judgment bias. Furthermore, considering the influence of personality traits on the JBT, we predicted that more anxious animals may show more pessimistic bias than highly sociable and confident individuals (as previously observed in dogs [14]). Finally, we investigated if the structure of the spatial JBT employed could have had limitations in its application in horses.

## 2. Materials and Methods

### 2.1. Ethics Declarations

The study was evaluated and approved by the scientific Ethics Committee for Animal Experimentation of the University of Bologna, the experiments were not invasive and were performed following regulations of Legislative Decree no. 26/2014.

### 2.2. Animals and Housing

The study involved eleven adult horses and five ponies from the same facility, to avoid variables due to different environmental conditions. To have a sufficient number of animals, we were not able to select for sex and/or breed. The sample comprised two stallions, eleven mares, and three geldings. The age of the subjects ranged from five to 25 years (mean age = 14 ± SD 6.6 years).

The overall population of 20 horses of the studied facility was housed in individual boxes (3.0 × 3.5 m), with daily access to individual outdoor paddocks during favorable weather conditions. Natural lighting was provided by windows, and the boxes were equipped with automatic waterers. Depending on their individual needs, the horses were fed with hay (7–10 kg per day) divided into three meals (at 7:30 a.m., 12:30 p.m., and 6:00 p.m.), supplemented by pelleted food with maize and carob (0.8–1.0 kg) in the evening meal. The center of the yard was occupied by two rectangular sand arenas (12 × 20 m and 15 × 50 m), a grass field (25 × 65 m), and a round sand pen (18 m diameter). All the horses were used for riding lessons for adults and children.

### 2.3. Personality Assessment

The owners and handlers filled out the online E-BARQ for each horse. This validated questionnaire is composed of 100 question items, divided into 13 categories for the ridden horse: Trainability (TR), Rideability (RID), Boldness (BOLD), Handling Compliance (HC), Working Compliance (WC), Easy to Stop (ETS), Human Social Confidence (HSC), Non-Human Social Confidence (NHSC), Novel Object Compliance (NOC), Touch Sensitivity (TS), Independence (IND), Easy to Load (ETL). Items indicating negative traits were scored on a 5-point scale, where Never = 5, Rarely = 4, Sometimes = 3, Usually = 2, Always = 1. A reversed scale was used for positive items, where Always = 5 and Never = 1. Therefore, a higher score in each category is always desirable. The total score for each category was automatically calculated upon completion of the online questionnaire (website: https://e-barq.com/ accessed on 25 November 2021) [31,32].

The ETL category was excluded during the data processing because most respondents did not answer its included questions.

### 2.4. Judgment Bias Test (JBT)

A Go/No-go spatial JBT has been employed, considering the protocol previously used by Freymond et al. [22] and Henry et al. [24]. To provide a familiar environment, one of the rectangular sand arenas (12 × 20 m) of the facility was used. Each horse was always handled by the same experimenter (exp. 1), who remained neutral and did not give any indication during the JBT to not interfere with the decision-making process of the horse. To prevent the horses from seeing their content, black plastic buckets were chosen. The JBT procedure occupied four days during the week: the first day was dedicated to the habituation, the second day to the training, and, after one day off, two consecutive days were devoted to the two sessions of the test.

On the first day, the subjects were familiarized with the setting. One subject at a time was conducted in the arena by exp. 1. In the arena, a start line was delimited by two red cones. A second experimenter (exp.2) positioned the bucket with a food reward on the ground in the positive position (P), 9 m from the start line (Figure 1). The habituation was completed when horses, released at the start line, independently approached the bucket three consecutive times. Half of the horses had the positive cue to their right and the others to their left. During the second day, horses learned to discriminate between the P and the opposite negative position (N), where the bucket was empty. N was 12 m from P (Figure 1). Each trial lasted a maximum of 60 seconds and P and N were presented one at a time in a pseudo-random order. No more than two consecutive trials of the same position occurred. For each horse, the latency to reach the bucket was recorded, and the subject succeeded in the training phase when the longest latency to reach the last three positive buckets was shorter than that of the three last negative ones [33]. The training phase was followed by a day off.

Two sessions of testing were performed on two consecutive days. Spread along a semicircle, three intermediate positions between N and P were presented one at a time: Near Negative (NN), Middle (M), and Near Positive (NP). In the intermediate positions, the bucket was empty. Each session followed the scheme proposed by Henry et al. [24], where ambiguous locations were preceded alternately by positive and negative locations (Test 1: P-N-NP-P-N-M-P-N-NN-P; Test 2: N-P-M-N-P-NN-N-P-NP) (Figure 1).

All positions were marked on the ground with small cones and were 9 m from the start line. The start line was in the north part of the arena for five subjects, while in the south part for the other five.

### 2.5. Video Recording and Analyses

Training and test sessions were videotaped using a Sony HDR-CX240E camera. The camera was placed on a tripod outside the arena beginning with the habituation phase to familiarize the horses with it. All videos were analyzed by the experimenters, previously trained, using the software Boris v. 7.9.19 [34], to record precisely the time needed by the horses to reach the buckets and to register their stress-related behaviors. The behavioral analysis was carried out using a specific ethogram [35,36,37,38,39,40] (Appendix A and the frequency (acts/minute) for stress behaviors was calculated.

### 2.6. Collection of Fecal and Horsehair Samples and Cortisol Assay

The fecal cortisol level was monitored for one week in each horse, while the only variation in their daily routine was the habituation and training for the test and the test procedure itself, four days out of the seven. For all 14 subjects considered in the JBT, feces were collected every day: on the habituation day, on the training day, on the day off between the training phase and the testing phase, on the two testing days, and on the two following days. This testing schedule has been chosen to reflect the HPA activation on the day before, as recommended by Möstl et al. [41].

A total number of 98 fecal samples (seven per horse) have been collected from physiological (spontaneous) defecation, about 24 h after the tests, usually in the morning.

Each fecal sample was placed in a labeled non-sterile plastic bag and frozen at −20 °C. The horsehair samples (*n* = 14) were collected from the base of the tail of each horse, except two, for which we used the mane. The horsehair was collected during the habituation day. All samples were stored at room temperature.

Cortisol concentrations were determined by radioimmunoassays (RIAs) based on the binding of 3H-steroid by competitive adsorption [42]. After filtration and lyophilization of the feces, the extraction methodology was modified considering Schatz and Palme [43]. Cortisol was extracted from 500 mg of feces with methanol–water solution (*v*/*v* 4:1) and ethyl ether. The portion of ether was vaporized under an airstream suction hood at 37 °C. The dry residue was finally dissolved again into 0.5 mL PBS (0.05 M, pH 7.5). The extraction of the cortisol from the hair was performed as described by Accorsi and colleagues [44]. Cortisol metabolite assay was carried out according to Tamanini et al. [45]. The cortisol RIA was performed using an antiserum to cortisol-21-hemisuccinate-BSA (anti-rabbit), at a working dilution of 1:20,000 and 3H-cortisol (30 pg/tube vial) as a tracer. Validation parameters of analysis were: sensitivity 0.19 pg/mg, intraassay variability 5.9%, interassay variability 8.7%. Radioactivity was determined using a liquid scintillation β counter and a linear standard curve, ad hoc designed by a software program [46].

All concentrations were expressed in pg/mg of hair and fecal matter.

### 2.7. Data Analysis

To avoid biases caused by differences in baseline running speeds, due to the size and/or age of individuals, raw latencies recorded to reach the intermediate positions were transformed into scores, according to the formula proposed by Mendl et al. [33]:$$\mathrm{Adjusted}\;\mathrm{latency}=\frac{\left(\mathrm{mean}\;\mathrm{latency}\;\mathrm{to}\;\mathrm{ambiguous}\;\mathrm{location}-\;\mathrm{mean}\;\mathrm{latency}\;\mathrm{to}\;P\right)}{\left(\mathrm{mean}\;\mathrm{latency}\;\mathrm{to}\;N-\mathrm{mean}\;\mathrm{latency}\;\mathrm{to}\;P\right)}\times 100.$$

This formula returns 0 for the P and 100 for N.

The whole statistical analysis was carried out in the RStudio environment [47].

The data distribution was assessed with the Shapiro–Wilk test. To compare the recorded scores in intermediate positions and fecal cortisol levels during the week, we conducted Friedman tests and pairwise Wilcoxon rank tests with post hoc Holm correction as. A Mann–Whitney U test was used to check if the latency to reach the intermediate positions was affected by P positions. Then, we compared the score obtained from E-BARQ of horses with P on their left and subjects with P on their right through a Mann–Whitney U test. Finally, we checked the Wilcoxon effect size (r) and 95% confidence intervals (CIs) for significant results.

To investigate the influence of horsehair cortisol levels on JBT results, a simple linear regression was employed, and different multiple linear regressions were used to model scores for each intermediate position as functions of (1) fecal cortisol levels and behavioral signals, (2) E-BARQ scores of personality categories. Collinearity was checked using multipanel scatterplots, Pearson correlation coefficients, and variance inflation factors (VIFs). We used backward selection based on the Akaike information criterion to find the optimal model. To verify the underlying assumptions of homoscedasticity, normality distribution, and independence of residuals, we employed the Breusch–Pagan, Shapiro–Wilk, and Durbin–Watson tests, respectively. When the model normalization was necessary, we used the Yeo–Johnson transformation, because of the presence of negative values.

The statistical significance was set at *p* < 0.05. Graphs were realized using the package ggplot.

## 3. Results

Two subjects were excluded from the statistical analysis because they failed the inclusion criteria of the JBT, and 10 horses and 4 ponies composed the final sample. Those that succeeded in reaching the criterion learned the task in an average of 16.5 trials (min 11–max 23).

According to the Friedman test, the latency of the subjects seemed to be significantly influenced by bucket position (χ^2^= 35.543, df = 4, *p*  <  0.0001). In particular, there was a significant difference between the latency to reach N (56.8 ± 8.51sec) and all the other positions (*p* < 0.01), while no significant difference was found between the latency to reach P (8.38 ± 1.27 s) and all the intermediate positions. According to the post hoc test, the subjects approached NN (13.6 ± 25.3 s) significantly slower than M (8.50 ± 0.62 s; *p* < 0.01, r = 0.88, CI (0.88, 0.89)). In addition, they approached NP in a mean of 8.76 ± 1.73 s (Figure 2).

Due to the variation of size and baseline running speed between the subjects, raw latencies recorded to reach the intermediate positions were transformed into scores. The position of the positive cues (P) seemed to have influenced the speed of horses to reach M and NN. The seven horses that had the P at their right side registered significantly lower scores to reach NN (right-P = 2.31 ± 8.33; left-P = 55.92 ± 39.40; W = 3, *p* < 0.01, r = −0.735, CI (−0.845, −0.38)), and a lower score in M with a significant tendency (right-P = −0.50 ± 2.36; left-P = −0.65 ± 10.80; W = 3, *p* = 0.07).

### 3.1. Influence of Stress on JBT Results

During the week of the test, cortisol concentration remained constant (Figure 3). Considering the multiple linear regression model, the possible acute stress induced by the entire test procedure and assessed through fecal cortisol and stress-related behaviors (Appendix A) did not significantly influence JBT results (NP, F (2,11) = 1.615, *p*= 0.24, adjusted-R^2^ = 0.09; M, F (2,11) = 1.83, *p* = 0.21, adjusted-R^2^ = 0.11; NN, F (2,11) = 1.073, *p* = 0.37, adjusted-R^2^ = 0.01).

In addition, we did not find significant influence of the chronic activation of HPA (Figure 4), assessed using horsehair cortisol concentration, on JBT results (NP, F (1,12) = 1.202, *p* = 0.29, adjusted-R^2^ = 0.01; M, F (1,12) = 0.008, *p* = 0.93, adjusted-R^2^ = −0.08; NN, F (1,12) = 0.95, *p* = 0.35, adjusted-R^2^ = −0.003).

### 3.2. Influence of Personality Traits on JBT Results

Comparing the E-BARQ scores of horses that have P on their right and those with P on their left, no significant difference was evidenced.

The analysis of the E-BARQ results (Figure 5) revealed collinearity between (1) TR and RID; (2) BOLD and NOC, HC, FG, NC; (3) HSC and TS. Therefore, for the model construction, we excluded RID, NOC, HC, WC, FG, and TS.

We perform multiple linear regression to model NP, M, and NN scores as a function of TR, BOLD, ETS, HSC, NHSC, and IND. Finally, we used backward selection based on the Akaike information criterion to find the optimal models.

The model that included HSC and IND was the best one (AIC = 38.58) to explain a significant amount of the NP score (F (2,11) = 7.919, *p* < 0.01, adjusted-R^2^ = 0.52). The model revealed that the NP score was positively predicted by IND (β = 7.740, t = 3.891, *p* = 0.002) and HSC, with a significant tendency (β = 3.477, t = 1.997, *p* = 0.07).

The model that included BOLD, ETS, HSC, and IND was the best one (AIC = 33.49) to explain a significant amount of the M score (F (4,9) = 4.962, *p* = 0.02, adjusted-R^2^ = 0.55). The model revealed that the M score was negatively predicted by ETS (β = −4.354, t = −2.953, *p* = 0.02) and positively predicted by BOLD, with a significant tendency (β = 2.572, t = 2.249, *p* = 0.05).

None of the personality categories seems to have influenced the score in NN.

## 4. Discussion

The first relevant result of our study underlines a possible bias due to the position of the positive cue that can influence the latency of horses towards the NN position, disturbing JBT results. Horses who have P on their right were faster to reach NN than horses with P on the left. One hypothesis can be the presence of environmental or sensorial disturbances near the arena that could be relevant for the horses but that humans do not notice or cannot control, such as particular odors, sounds, or the presence of other subjects [48] in the stalls that were near one side of the arena. However, we randomized the position of P and, as specified in the Materials and Methods section, we positioned the buckets in the north part of the arena for nine horses, while for the other five we positioned the buckets in the south part. The second hypothesis was the presence of the phenomenon of lateralization that can cause biases in motor behavior or perception of stimuli [49]. In the setting of tests that employ spatial tasks, laterality should be considered, especially for prey animals, because lateralization can influence their performance. In particular, studies show that horses usually prefer the use of the left eye to evaluate and explore a new environment and novel object with an emotional valence, both positive and negative [27,50]. The use of the left eye is associated with rapid reaction [28] and with both positive and negative associations, while the right eye is used to examine neutral objects [27]. We cannot exclude that the presence of NN to the left of the horse has possibly led to a different evaluation of the intermediate position. We think that further research is necessary to explore lateralization as a possible source of disturbance in the horse spatial JBT.

According to statistical analysis, there was no difference in latency to reach P and ambiguous locations, even if the latency to reach NN recorded a high variability within the group and was different compared to those in front of NP and M. NP and NN often are perceived as P and N training cues by animals. However, the value associated with P and N can change the animal’s decision to take a risk. Therefore, the lack of a reward in the N position, as in our case, is not so costly for the individual, and it can also decide to respond positively in front of a potential negative cue [7,10].

Similar to our results, Müller and colleagues [51] found that half of their dog population had similar latencies in approaching NP and M, but went slower to NN, whilst the other half approach the ambiguous locations from NP to NN increasingly slowly. They justified this discrepancy with a difference in personality traits [51]. Indeed, in our study we found that some of the horses’ personality categories influenced the response toward NP and M. However, we did not find any relationship between the E-BARQ scores and latency to reach NN. In particular, horses with higher scores in IND and HSC, with a significant tendency, approach NP slower, whilst the subjects with lower scores in BOLD, with a significant tendency, and higher scores in ETS approach M faster. The traits IND and BOLD are general categories including anxiety, separation anxiety towards conspecifics, and avoidance behavior in a novel situation. Therefore, low scores in these two personality traits can indicate an anxious and fearful subject [28]. Considering that, our results disagree with previous studies, where animals characterized by anxiety, fear, and aggression traits approach intermediate positions slowly, expecting a negative outcome, and show a pessimistic bias [15,30,52,53]. However, our results could have been influenced by several factors, such as the small sample size and the lack of homogeneity for sex and breed in the sampled population, the spatial nature of the test (as stated previously), and the fact that the horses involved in the test, also for safety reasons, were not aggressive subjects. The other two results are more easily justifiable. The fact that subjects with lower HSC scores reached the NP faster can be justified by the fact that horses that are less confident towards humans may prefer to increase the distance from the handler, especially in front of the NP cue, which often is associated with P and, thus, is associated with a high expectation of a reward [10]. Finally, the trait ETS considers how the horse responds to human signals to stop. Therefore, we cannot exclude that those horses with higher ETS tend to respond better to human signals, so when they were released, they immediately started to approach M. Considering this last result, even if in this study we tried to avoid possible human influences and interactions with horses, future applications of this test should take this factor into account.

The effect of personality traits on JBT is difficult to explore because this test can also be influenced by both long-term mood and stressful or rewarded recent experiences [4,54,55,56]. We employed horsehair and fecal cortisol to investigate chronic and subacute HPA activation, and to evidence particular stressful situations. Cortisol is a useful tool to investigate animal welfare [6], but on its own it is not exhaustive of animal emotion and the welfare status. In particular, fecal cortisol concentration represents an integrated average of the circulating hormone over a species-specific amount of time, which in horses is about 24 h [29,41,57,58], and fecal sampling is non-invasive. On the contrary, plasma and saliva cortisol levels are prone to circadian patterns and short-term fluctuations, represent point-of-time estimates, and can be altered by the sampling procedures, which can cause stress in animals that are not trained for them [41,57,58]. Therefore, collecting fecal samples from each subject throughout the entire week allowed us to monitor the stress level of animals possibly due to the entire procedure of the JBT, which could have represented a disruption of their daily routine, and to compare cortisol levels during the test procedure and in the baseline condition. We did not record significant differences in fecal cortisol concentration during the days of the test procedure. In addition, both fecal cortisol concentration and relative frequency of stress behaviors seemed to not influence JBT results. Therefore, the test itself probably did not induce particular stress in the studied subjects, but it should be considered that testing social animals, such as horses, individually for the JBT could increase stress [59]. This aspect should always be considered because, to avoid biases in the decision-making process of the animal, it is important that habituation, training, and testing phases do not cause stress.

Considering horsehair cortisol, this represents the chronic activation of HPA and can be a potential indicator of welfare and long-term stress [29]. Indeed, cortisol is continuously incorporated into the hair during its growth. The cortisol concentration in this matrix can be a useful indicator of the HPA activity during the weeks preceding the sampling [60]. Each individual, also according to different personality traits, copes differently in the same environment, and in a group of animals that came from the same condition, we can find different levels of chronic stress [61]. This has been well represented in our sample population, where each individual presents a different concentration of horsehair cortisol. Our prediction was to find a positive relationship between horsehair cortisol level and ambiguous position score, finding that subjects with higher horsehair cortisol concentration show a more pessimistic bias. However, this hypothesis has been refuted and this aspect should be deepened by further research, enlarging the sample size, using a multidisciplinary approach in horse welfare evaluation, and refining the JBT structure considering horses’ characteristics.

## 5. Conclusions

Considerations of personality traits and cortisol levels in horses’ JBTs are the innovative aspects of this research. However, this preliminary study evidenced some limitations of the use of spatial Go/No go JBT tasks in horses. In particular, the spatial nature of this test and individual personality traits can influence the responses of horses to intermediate positions and alter the interpretation of the JBT. Further research is needed to find a standardized JBT for horses that takes into account setting, species peculiarities, and individual motivations. Finally, future application of the JBT to assess affective states in horses should check for stressful situations and personality traits.

## Figures and Tables

**Figure 1 animals-12-03014-f001:**
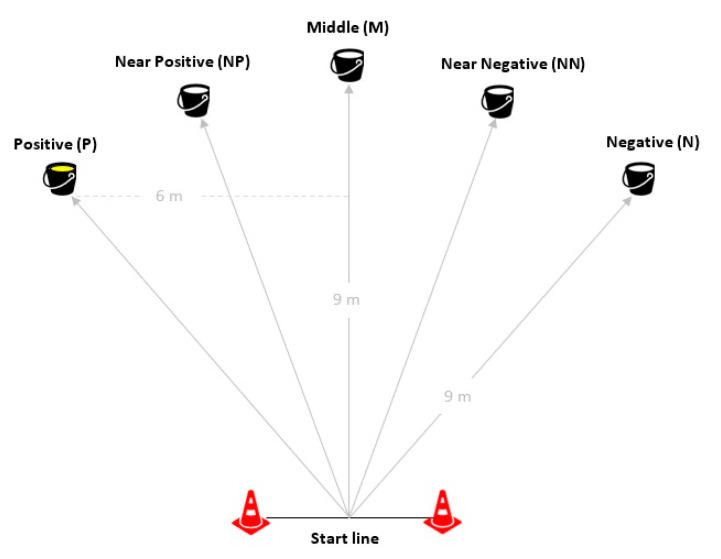
The setting of the spatial judgement test. P is the Positive position (bucket with food) and N is the Negative position (empty bucket). The ambiguous positions (empty bucket) are Near Positive (NP), Middle (M), and Near Negative (NN).

**Figure 2 animals-12-03014-f002:**
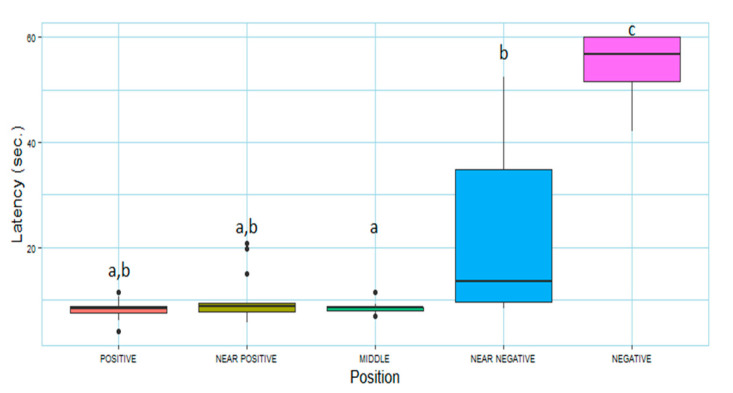
Latency in seconds (sec.) to reach the bucket in the five locations. Results refer to the two days of the test. The bar within the box represents the median, the borders of the box are upper and lower quartiles, the bottom and top whiskers signify the lowest and highest cases within 1.5 times the interquartile range (IQR), and outliers are shown through black full circles. a, b, and c indicate significant results (*p* < 0.05).

**Figure 3 animals-12-03014-f003:**
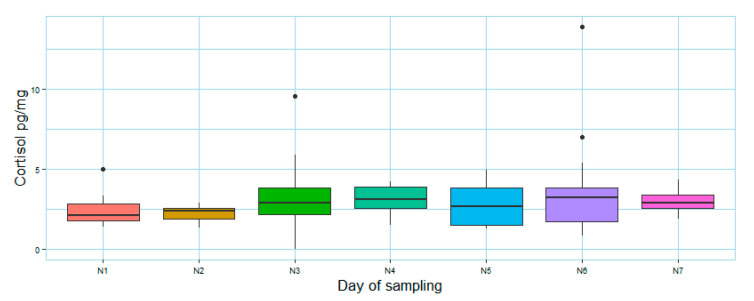
Fecal cortisol (pg/mg) measured from samples collected every day of the JBT week (N1 = habituation, N2 = training, N3 = resting day, N4 = test1, N5 = test2, N6 = resting day after the last test day, N7= second resting day after the last test day). The bar within the box represents the median, the borders of the box are upper and lower quartiles, the bottom and top whiskers signify the lowest and highest cases within 1.5 times the IQR, and outliers are shown through black full circles.

**Figure 4 animals-12-03014-f004:**
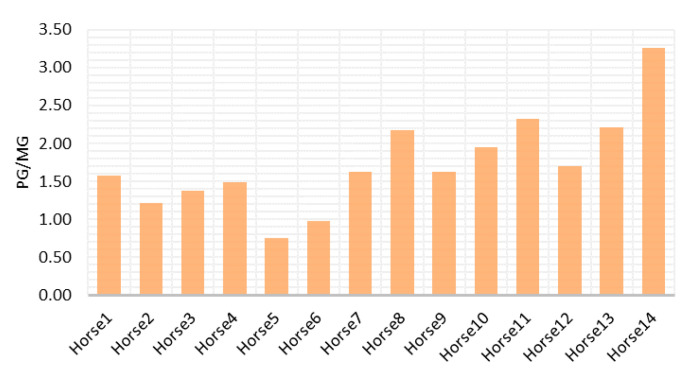
Cortisol level in horsehair (pg/mg) of each subject considered in the research.

**Figure 5 animals-12-03014-f005:**
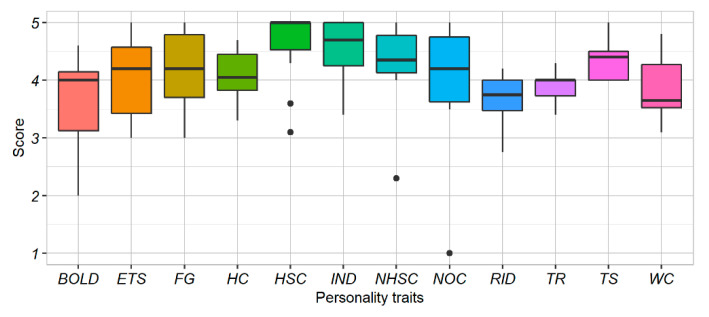
E-BARQ score for each personality trait. The box plots show the median and 25th and 75th percentiles; the whiskers indicate the values within 1.5 times the interquartile range, IQR. The outliers are indicated by the black circles.

## Data Availability

Data available on request.

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
