# Peer review of "Limitations of Spatial Judgment Bias Test Application in Horses (Equus ferus caballus)"

_animals, 2022, doi:10.3390/ani12213014_

Round 1
Reviewer 1 Report
Limitations of spatial judgement bias test application in horses (Equus ferus caballus)
General comments: Interesting study worth publishing. However, the quality of the article is poor. Need to do substantial revision in terms of presentation. Methodologies are described in the results section. References are not in proper format.
Line 21: Deepening these aspects ……
Deepening what aspects? Sentence restructure
Line 34 : This preliminary study evidences…
Sentence restructure. This preliminary study provides evidence that there are limitations in the employment of spatial JBT….
Line 49: Not a good way to start a paragraph with “Therefore”
Line 49: “More and more”. Use more specific words
Line 51: arousal4. I presume “arousal (4)”
Line 121 - 123: Previous studies on JBT (which you have cited) have shown that sex is an factor which affects outcome. Why did you keep mixed sexes in the sample? Please explain. Also breed is closely associated with personality straits. What is the breed of your sample animals?
Line 156-157: Horses were handled always by the same person, who did not give any indication during the JBT.
This sentence is not clear. What type of indication are you talking about? You mean leading the horses to the correct bucket which is not good.
Line 290: Each horse “included” in the research….. Not “considered”.
Line 292-307: Most of this should be in the “method section”. Only discuss results in the results section.
Line 438: Is reference number 18 missing? Maybe you have it after 17. Please follow the correct format for all references.
Example: Iigaya, K.; Yi, S.; Wahle, I.A.; Tanwisuth, K.; O’Doherty, J.P. Aesthetic Preference for Art Can Be Predicted from a Mixture of Low- and High-Level Visual Features. Nat. Hum. Behav. 2021, 5, 743–755.
Author Response
We wish to thank the Reviewer for their thoughtful comments and suggestion. We have revised the manuscript presentation and the format of references. The English has been edited and revised by a mother language speaker. We believe that the revised version of the manuscript clarifies some points of uncertainty and provides more information to the readers. Please see the attached pdf for the point-by-point response.

Reviewer 2 Report
Dear authors, thank you so much for your manuscript. It was a pleasure for me to review it. I have some comments that I hope can help to improve the quality of your manuscript.
Title – the title should reflect the aim of your study. First of all, you should “standardize” your aim throughout the simple summary, abstract and introduction.
Abstract
Line 26 – 14 horses? Or 11 horses and 5 ponies (Line 125)?
Lines 31 and 32 – Positive should be “Near Positive”? It is not clear for me is the three ambiguous positions (Near Positive, Middle, Near Negative) were characterised by the bucket with the food reward
Introduction
Line 98 – I suggest adding here a short paragraph in which you can introduce the concept of lateralisation in horses. Accordingly, I may suggest reading this paper “Vinassa et al. 2020 Palatability assessment in horses in relation to lateralization and temperament, Applied Animal Behaviour Science 232, 105110 https://doi.org/10.1016/j.applanim.2020.105110”
Line 51 – delate 4 after arousal
Materials and Methods
Line 153 – Can you explain a little bit more about how you have modified the spatial JBT of Freymond and Henry? So your methodology can be repeatable.
Line 154 – it is 12m x 20m or 12m x 25m (as stated in line 134?)
Line 155 – where the buckets were placed?
Lines 156-157 – it is a sentence that I do not understand…please reword... which person? Indication about what?
Line 159 – Were adapted to the setting
Line 161 – It is useful to refer also here to Figure 1.
Line 161 – released at the start line … where the start line was? Could be useful to add the meters
Line 183 – where the camera was placed? Were the animals adapted also to the presence of the camera?
Line 184 – add the version of the software
Line 185-187 – who analysed the videos?
Results
- according to line 125 you have only 11 horses …
Generally, results are well presented
Discussion
The discussion is well written underlining the limitations of the study and discussing the results on the basis of the current literature. I have just two minor comments
Line 323-324 – this aspect need to be clarified in the m&m
Line 335-337 – I suggest to delete this sentence
Author Response
We wish to thank the Reviewer for their thoughtful comments and suggestion. We believe that the revised version of the manuscript clarifies some points of uncertainty and provides more information to the readers. Please the attached pdf for the point-by-point response

Reviewer 3 Report
Because British and American English was mixed in the paper, with the prevalence of American English, my suggestions/corrections respect the latter.
Please check the Instructions for Authors and remove the empty lines (or spacing before/after lines) where these are not needed.
The paper starts well and promising, but it falls flat later, especially in the Discussion and Conclusions sections. The English language needs extensive editing, because part of the unclarities may be caused by incorrect phrasing. However, the first part seems to be written (or edited) by a different person than the rest of the paper, the Introduction’s writing is better. Please find below my suggestions, comments, and questions to be used as considered to improve the quality of the manuscript.
Simple summary
L11: consider changing to a synonym one ‘evaluate’. Please use past simple tense when describing own research (‘aimed’ instead of ‘aims’ here) throughout the paper
L13: please change ‘animals’ to ‘their’. Although the terms ‘horse’ and ‘animal’ can often be used interchangeably to avoid repetition, the paper describes a study in horses and not generally in animals
L14: ‘a’ before ‘food’ can be deleted t reduce the word count
L15: consider changing ‘animals’ to ‘the subjects’
L17: please delete ‘last’, it is redundant. Please change ‘allows us to define them’ to ‘allowed their classification’
L18: ‘showed’ instead of ‘shows’ (please use simple past n describing own research throughout the paper)
L19: please change ‘animals’ to ‘horses’—avoid generalizing your results
L19: you may wish to change ‘this kind of test’ to ‘such tests’ (word count reduction—WCR)
L20: ‘to be’ can be deleted (WCR); consider changing ‘show’ to ‘use’ or similar
L21:’suggested’; ‘type’ instead of ‘kind’
L22: ‘in horses’ (or simply ‘in horse welfare’ instead of ‘of horses’
Abstract
L25: ‘aimed’
L26: numbers beginning sentences should be spelled out. However, avoiding numbers as sentence beginners is preferred. I would suggest ‘The distinction between two positions […] was taught to 14 horses’. To avoid personal voice, the sentence could continue with ‘[…] then the latency to measure three ambiguous positions […] was measured’
L29: change the word order to ‘using the validated E-BARQ questionnaire.’ E-BARQ: each abbreviation has to be spelled out at their first usage
L31: ‘showed’
L32: ‘some’ is a vague adjective, consider to change to ‘several’ or an exact number. Please use ‘influenced’
L34: consider to use ‘highlighted’ or similar, instead of ‘evidences’; ‘some’ is a vague descriptor
L37: both of the species and of individuals
Introduction
L44: please insert ‘The’ before ‘Treaty’
L51: please correct ‘arrousal4’
L57: please change ‘this’ to ‘those’
L57-58: consider changing to ‘the emotions of animals’
L59: give the spelled-out variant first (no capital H needed), then the abbreviation in brackets (HPA). The word ‘only’ is unclear where it is, is it necessary?
L61: please do not use contractions (doesn’t) in academic text. Consider inserting ‘the underlying’ before ‘emotion’ for more clarity
L62: I would prefer ‘indicators’ instead of ‘indexes’
L64: I would prefer ‘be incongruent’ or similar instead of ‘disagree’ (not an active verb)
L64: ‘it should be considered that’ can be deleted
L65: please consider ‘facing’ instead of ‘in front of’ (fluency)
L65: ‘non-verbal animals’ is redundant. You could change to ‘in the non-verbal context of animals’ or similar, or delete ‘non-verbal’
L68: ‘surely’ is not needed. Do you mean ‘surrogate’ by ‘proxy’? Please consider to change to a more academic synonym
L70: consider changing ‘so’ to ‘and thus,’
L79: ‘one’ is not needed
L80: consider changing ‘in front of’ to ‘when facing’ or similar; ‘consists of’ to ‘includes’ or ‘implies’/’involves’
L81: insert ‘the’ before ‘animals’
L82: insert ‘the’ before ‘animals’. To avoid repetition, consider changing to ‘Once the animals have learned the difference, they are presented to ambiguous stimuli’. Delete ‘some’, use present tense (not own research result, describes a method)
L83: ‘usually’ is vague, please be more precise
L92: insert ‘the’ before ‘decision-making’
L93-97: grammatically correct, but for better fluency keep tense consistency in the sentence and paragraph: consider changing to present the simple past (as used right after ‘it was found’)
L98: ‘indicate’ (present tense) or similar, instead of ‘pointed out’
L99: delete ‘the’ before ‘interaction’
L100-101: consider changing to ‘However, the research of JBT in horses broadened only recently, especially as regards the effect of the environment and training method on the affective state of this species’
L102-104: consider changing to ‘Nevertheless, further research is warranted because of the lack of a standardized protocol and insufficient studies on the influence of personality traits on JBT results in horses’
L105: change ‘aims’ to ‘aimed’
L105-117 consider improving fluency, providing the spelled-out version before the abbreviation, and avoiding repetition of ‘we’. Suggested phrasing: This research aimed to investigate the application of a spatial JBT in horses, considering the influence of their stress level and personality traits, and the test structure itself. For this, we explored the acute and chronic activation of the HPA axis (by measuring fecal and horsehair cortisol levels [23-24]) and assessed the horses’ personalities (by the Equine Behaviour Assessment and Research Questionnaire—E-BARQ [25]). Our hypothesis was that both acute stress, potentially caused by the test procedure itself, and individual chronic stress could influence the horses’ responses during the JBT test causing a negative judgment bias. Furthermore, considering the influence of personality traits on JBT, we predicted that more anxious animals may show more pessimistic bias than highly sociable and confident individuals (as previously observed in dogs [14]). Finally, we investigated if the structure of the spatial JBT employed could have had limitations in its application in horses.’
Please note that citing references within the study’s aim is rather unusual, but I do not consider it impermissible.
Materials and methods
L125: to avoid a number at the sentence beginning, it could be rephrased as ‘The study involved eleven […]’
L126: numbers lower than 10 have to be spelled out (3)
L128-and on: as the study was already performed, please change its description to simple past tense
L129: change the word order to ‘had daily access’
L128-137: please improve fluency and style. Also, please use consecrated symbols (× instead of x) Suggested: ‘The overall population of 20 horses of the studied stable were housed in individual boxes (3.0 × 3.5 m), with daily access to individual outdoor paddocks during favorable weather conditions. Natural lighting was provided by windows, and the boxes were equipped with automatic waterers. Depending on their individual needs the horses were fed with hay (7-10 kg per day) divided into three means (at 7:30 AM, 12:30 PM, and 6:00 PM), supplemented by concentrates (0.8 – 1.0 kg) at the evening meal. The center of the yard was occupied by two rectangular sand arenas (12 × 25 m and 15 × 50 m), a grass field (25 × 65 m), and a sand rod (?) (18 m diameter). The facility had a clubhouse, a saddlery, and service rooms. All the horses were used for riding lessons (especially fieldwork) for adults and children.’
What was the concentrate feed composed of?
What do you mean by ‘rod’?
The clubhouse, saddlery, and service rooms don’t seem relevant to the study, please exclude.
What do you mean by ‘fieldwork’ (horse usage for riding)?
L139: spelling out is not necessary anymore
L140-151: please reverse the order of the paragraphs, to improve the reader’s understanding. After the 1st sentence of the subchapter include the description of the E-BARQ (L147-151). Please complete this description, describe concisely but in more details the protocol, thus the reader would understand the paper in itself, without compulsory additional documentation on the E-BARQ. L148: change to ‘were scored’. What is Likert? Please indicate. L150: insert ‘the’ or ‘a’ before ‘higher’.
Then continue with the paragraph L142-146. Consider changing ‘we did not consider the category “Easy to load” because […]’ to ‘The “Easy to load” category was excluded during the data processing because most respondents did not answer its included questions.’
L153-187: to me the fragmentation of this segment seems excessive. I would suggest putting all this content under a single subtitle, for example, ‘Judgement bias testing’.
Language fluency and style should be improved, please find a few suggestions below.
L153-157: suggested change: ‘For this aspect of the study a spatial JBT modified by Freymond et al. [19] and Henry et al. [21] was applied by two researchers. To provide a familiar environment one of the facility’ s rectangular sand arenas was used (12×20m), and the horses have been always handled by the same person, who did not give any indication during the JBT. To prevent the horses from seeing their content, black plastic buckets have been chosen.’
Issues: please check the Instructions for authors for writing the dimensions, and be consistent (in the previous subsection you inserted spaces before and after the multiplication sign, but here you did not). Please use the multiplication sign and not the ‘x’ letter. What is the correct dimension of the arena (you gave a different dimension in the previous subsection)?
L159: Suggested: ‘On the first day the horses’
L164: insert ‘the’ before ‘horses’
L165-166: suggested: ‘maximum 60 seconds’
L172-181: for a better repeatability of the study please mention the distance between the start line and the buckets
L183: insert ‘device’ or ‘camera’ or similar, after HDR-CX240E
L184-185: suggested: ‘All videos were analyzed using the Boris software, to record precisely the time needed by the horses to reach the buckets, and to register their stress-related behaviors.’ Please state in brackets the source of the software.
L186: ‘the frequency’
L189: suggested: ‘A total number of 98 fecal samples (seven per horse) have been collected from physiological (spontaneous) defecation, about 24 hours after the tests, usually in the morning of the resting day.’
L192: ‘bag’ instead of ‘bags’; frozen at -20; insert ‘The’ before ‘horsehair’
L195: insert ‘The’ before ‘cortisol’
L195-208: please move the second sentence after the description of the testing methodologies. Suggested: ‘[…] the modified Schatz and Palme [36] methodology’ Please explain briefly what modification you made. Please remove the publication years, those are included in the reference list and not needed in the text (here and throughout the paper)
L209-216: please move the sentence in line 206 to the beginning of this subchapter
L218: suggested: ‘The data distribution was assessed with the S-W test.’
L222: ‘compared’ instead of ‘compare’
L226: suggested: ‘To investigate the […] a simple linear regression was employed, and different multiple …’
L229: suggested: ‘collinearity was checked using…
L233: delete ‘respectively’ (or move it at the end of the sentence)
L237: suggested: ‘The statistical significance was set at p<0.05’
Results
L242: insert ‘the’ after ‘according to’ and consider to use ‘the latency of the horses’
L244: delete ‘employed’
L245: consider to change ‘there was no significant difference’ to ‘no significant difference was found/noticed/observed/recorded’
L247: ‘the horses approached’
L248: ‘they approached […] significantly faster than NN’–no tendency calculation was performed
Figure 2: all abbreviations have to be spelled out at first usage, what is IQR?
Please revise the English of the paper. The use of articles and prepositions, the fluency, word order in sentences, and conciseness have to be corrected. From here on I will copy-edit only the more serious issues in this regard.
L256: what do you mean by baseline running speed?
L267-267: please move this sentence to the Materials and methods section. Also, this description seems different from what it can be understood in that section about the feces sampling procedure.
L253-276: How could you assess the influence of the intermediate positions on the fecal cortisol concentrations of the horses? More explicit descriptions in the Materials and methods section may be needed.
Fig 4. Is the mean concentration represented here? Please specify, or complete the Materials and methods section, if the samples were only once collected (and when?)
L239: ‘any’ is not a proper word here
L294-300: please rephrase this sentence for more clarity. Please consider using ‘excluded’ instead of ‘dropped’
Discussion
L322-323: What kind of disturbances do you mean here and on what basis do you consider these could have been relevant for the horses (citation would be welcome)? Please be more specific
L327: the ‘performance’ word seems senseless, please delete or reword the sentence
L332: suggested: ‘while the right eye is used to examine neutral objects’
L252-352: unclear what the ‘tendency’ refers to, please rephrase.
L355: what do you mean by ‘general’ category?
L357-358: please explain more on how you reached this conclusion and how the pessimistic bias is displayed
L360-361: what could have been a bias and due to what circumstance?
L362: please rephrase, what do you mean by faster horses?
L362-365: the relevance of the citation is not clear, and there is a contradiction between the statement above (that the horses were familiar with handling—and thus a bias), then here they would be avoidant toward the handler…
L368-370: here it sounds like a research recommendation for studying what you have decided to avoid to study. Rephrasing could give more clarity
L377-378: scientifically unsound statement. Same for L383-384. A huge body of research investigated cortisol variations in different mediums (blood, saliva, urine, faces, hair, etc), over time, in relation to stress in different situations, in horses. How fast, after the occurrence of acute stress, cortisol increases, for how long it stays elevated, and how much it increases? How are cortisol levels during chronic stress in different samples? This section needs a wider approach, using the available research results to explain those of this study. Please rewrite completely this section, or remove cortisol testing from the paper. If no result is relevant, then the testing is not relevant for the study—previous documentation should have clarified this.
The Discussions section fails in explaining the study results—it should be rewritten.
Conclusions
Very general and flat. Please be more specific.
Reference list
Please respect the Animal Journal template and re-format this list accordingly.
Author Response
We wish to thank the Reviewer for their thoughtful comments and suggestion. We believe that the revised version of the manuscript clarifies some points of uncertainty and provides more information to the readers. We have extensively revised the English and format of the article. The English has been edited and revised by a mother language speaker. Sorry, however, we would like to specify that the entire manuscript was previously written and edited by the same person. Please see the attached pdf for the point-by-point response

Round 2
Reviewer 3 Report
Dear Authors,
The readability and style of your article improved, and its scientific value increased following your editing. Two of my following suggestions, I apologize, I should have noted during the first review. Besides these, there are only two important aspects that still have to be addressed in the Materials and methods section (specifying the moment of horsehair sample collection and clarifying the exact sequence of study days—during the horses’ training, resting, and testing). The rest of my observations are suggestions for minor changes, to be used as considered, please find them below:
Abstract
L28-30: when abbreviations are spelled out at their first usage, the abbreviation should be in brackets. In the case of E-BARQ you can delete this way ‘questionnaire’, currently used twice (‘…the validated
Equine Behavior Assessment and Research Questionnaire (E-BARQ) was …’). By using a synonym, the repetition of ‘to employ’ should be avoided across these two sentences
L33-35: replacing ‘several’ with ‘certain’ in one of the two places could be beneficial
Materials and methods
L124: please be consistent in formatting the subtitles: Animals and Housing is the only one with capitalized words (I apologize for missing this in the first review round)
L167: consider deleting ‘distant’
L169: this might have been missed in the first review round: please do not begin the sentence with a number (replace ‘50% of horses’ with ‘half of the horses’
L178: consider changing ‘Following a semicircular’ to ‘Spread along a semicircular’
L193-194: for improved fluency, I would suggest ‘The camera have been placed on a tripod outside the arena beginning with the habituation phase to familiarize the horses with it.’
L200-204: the number of days with different activities is not clear enough, this has to be addressed. I give you a suggestion of the text as I understand the time-frames, but please change it as appropriate: it is important for the reader to understand clearly how many days were used for training, for resting the horses, and for testing. Another idea, maybe even better, would be to clearly state this schedule somewhere at the beginning of the Materials and Methods, maybe at the end of the ‘Animals and housing’ sub-chapter. However, consider my suggestion and correct it as appropriate: ‘The fecal cortisol level was monitored for one week in each horse, while the only variation in their daily routine was the training for the test and the test procedure itself, four days out of the seven. For all 14 subjects considered in the JBT, feces were collected every day: on the two training days, on the day off between the training phase and the testing phase, on the two testing days, and on the two following days.’ The following sentence, ‘Cortisol concentration in feces reflects the HPA activation of the day before [41]’ as it is, belongs to the Discussion section. In order to keep it here I would suggest ‘This testing schedule has been chosen to reflect the HPA activation on the day before, as recommended by Möstl et al. [41].’
L208: please state when were the horsehair samples collected within the study protocol
Discussion
Thank you for a much-improved Discussion section; congratulations for increasing considerably the value of your manuscript.
Author Response
Thank you for your suggestions. We hope that the manuscript has been improved and that these points have been clarified.
